# Contribution of ROS and metabolic status to neonatal and adult CD8+ T cell activation

José Antonio Sánchez-Villanueva[1], Otoniel Rodríguez-Jorge[1], Oscar Ramírez-Pliego[1], Gabriela Rosas Salgado[2], Wassim Abou-Jaoudé[3], Céline Hernandez[3], Aurélien Naldi[3], Denis Thieffry[3]*, María Angélica Santana[1]*

1 Centro de Investigación en Dinámica Celular (IICBA), Universidad Autónoma del Estado de Morelos (UAEM), Cuernavaca, Morelos, México, 2 Facultad de Medicina, Universidad Autónoma del Estado de Morelos (UAEM), Cuernavaca, Morelos, México, 3 Département de Biologie, Institut de Biologie de l'École Normale Supérieure (IBENS), École Normale Supérieure, CNRS, INSERM, Université PSL, Paris, France

* santana@uaem.mx (MAS); thieffry@ens.fr (DT)

**Data Availability Statement:** All relevant data are within the manuscript and supporting information.

## Abstract

In neonatal T cells, a low response to infection contributes to a high incidence of morbidity and mortality of neonates. Here we have evaluated the impact of the cytoplasmic and mitochondrial levels of Reactive Oxygen Species of adult and neonatal CD8+ T cells on their activation potential. We have also constructed a logical model connecting metabolism and ROS with T cell signaling. Our model indicates the interplay between antigen recognition, ROS and metabolic status in T cell responses. This model displays alternative stable states corresponding to different cell fates, i.e. quiescent, activated and anergic states, depending on ROS levels. Stochastic simulations with this model further indicate that differences in ROS status at the cell population level contribute to the lower activation rate of neonatal, compared to adult, CD8+ T cells upon TCR engagement. These results are relevant for neonatal health care. Our model can serve to analyze the impact of metabolic shift during cancer in which, similar to neonatal cells, a high glycolytic rate and low concentrations of glutamine and arginine promote tumor tolerance.

## Introduction

Infections in children under six months cause approximately four million deaths per year [1]. Neonatal T cells have low, tolerant or skewed responses, and a relatively high threshold of activation, potentially involving epigenetic mechanisms [2–5]. Compounds present in neonatal serum contribute to the low response of neonatal cells, among them adenosine [6, 7], arginase [8], and other proteins [9, 10]. Arginase and adenosine are metabolic inhibitors associated with a lack of T cell responsiveness in cancer [11–14]. Indeed, the pathways of arginine and glutamine metabolism are implicated in tolerance to tumors and are considered as therapeutic targets [15–17].

CD8+ T cells from newborns have distinct transcriptional and epigenetic profiles biased towards innate immunity and, albeit in homeostatic proliferation, their clonal expansion and effector functions are diminished [18]. Lower mitochondrial mass and membrane potential, as

The model is in the ginsim repository with number (http://ginsim.org/node/229)

**Funding:** CONACYT Grants 168182 and 257188 and the ECOS/ANUIES/SEP/CONACYT grants M11S01 and M17S02. The D.T. laboratory was supported by grants from the French Plan Cancer, in the context of the projects CoMET (2014–2017) and SYSTAIM (2015–2019), as well as by a grant from the French Agence Nationale pour la Recherche, in the context of the project TMod (2016–2020).

**Competing interests:** The authors have declared that no competing interests exist.

well as differences in calcium fluxes and in calcium and potassium channels have been reported [19–21]. Naïve, memory and effector functions depend on different metabolic programs, adapted to cellular requirements. Naïve T cells are metabolically dormant (quiescent). After an encounter with an antigen, they may either tolerate the antigen, become activated, or turn anergic. Despite important advances in the field of immunometabolism, a comprehensive view of the interplay between antigen recognition, metabolic status and ROS is missing. Our study combines experimental measurements with computational modeling to assess the role of metabolism, in particular of ROS, on T cell signaling.

We evaluated cytosolic (c) and mitochondrial (m) ROS levels on naïve CD8⁺ T cells from human newborns and adults, at basal level and after TCR stimulation. Our results suggest fundamental differences in the ROS signaling and redox status between CD8⁺ T cells from newborns and adults. Alterations of redox and metabolic nodes in the model result in low neonatal CD8⁺ T cells response, establishing the involvement of the metabolic status of the neonatal cells in their impaired response. The effect of ROS in cell function has changed from a waste product to an important signaling messenger [22]. Furthermore, the manipulation of ROS is being considered for cancer immunotherapy [23].

Using a logical formalism implemented in the software GINsim ([24] http://ginsim.org), we defined a comprehensive dynamical model integrating the most relevant signaling, metabolic and transcriptional regulatory components controlling the activation of CD8⁺ T lymphocytes.

This model displays alternative stable states, which corresponds to different cell fates, i.e. quiescence, activation and anergy, depending on ROS status. Stochastic simulations further suggest that the lower activation rate of neonatal compared to adult CD8⁺ T cells upon TCR engagement is attributable to differences in ROS status or glutaminolysis at the cell population level.

## Materials and methods

### Ethics statement

The collection of cord blood samples, with informed mothers' consent, was granted for this work by "*Comité de Etica en Investigación Hospital General de Cuernavaca Dr. José G. Parres. CONBIOÉTICA-17-CEI-001-20160329*". The collection of adult blood was granted by the ethics committee of "*Secretaría de Salud. Dirección de Atención Médica Departamento Centro Estatal de laTransfusión Sanguínea Oficio No*: *SH/CETS/438/2016*".

### Blood collection and cell purification

Cord blood was collected at *Hospital General de Cuernavaca Dr. José G. Parres*, with informed mothers' consent. Adult samples were obtained from leukocyte concentrates at the *Centro Estatal de Transfusión Sanguínea*. For each comparison, the minimum number of samples included in the neonate vs adult groups was three, for some cases the number of samples was up to nine. The total number of samples used for this study was 30 samples for neonatal CD8⁺ T cells and 30 for adult cells. No eligibility criteria were defined regarding the donors' gender. This parameter was distributed randomly across the samples and equivalent numbers of male and female donors were used for neonatal and adult cells. No differences related to gender were detected. For adult samples, the ages of the donors were between 26 and 40 years. All cord blood samples were obtained from vaginal deliveries and no drugs known to affect T cells responses were administered to the mothers during labour. All the samples were immediately processed after collection. CD8⁺ T cells were purified as previously described [25]. Briefly, the blood was centrifuged on LymphoprepTM (Axis-Shield, UK) and Mononuclear cells were incubated with 1 mL of erythrocytes and the RossetteSep^TM CD8⁺ T cell enrichment cocktail

to obtain Total CD8+ T cells (15063; StemCell Technologies, Canada). The memory cells were eliminated using magnetic beads (8803; Pierce; Thermo Fisher Scientific, Bremen, Germany) loaded with a CD45RO-specific mAb (eBiosciences, San Diego, CA). We obtained ≥94% naïve CD8+ T cells. The total elapsed time from blood collection to CD8+ T cells purification was similar for adults and neonatal samples, 46 to 48 hours.

## Cell stimulation

Naïve CD8+ T cells were cultured in RPMI containing 1% L-glutamine, antibiotics (100 U/mL penicillin and 100 μg/mL streptomycin) and 5% fetal calf serum. For stimulation 1 μg/mL or each, anti-CD3/anti-CD28 mAbs was used (OKT3, 70-0037-U100, CD28.2, 70-0289-U100, Tonbo Biosciences), cross-linked with anti-mouse mAb (405301, BioLegend). Cells were incubated for 6 h under 5% $CO_2$ at 37˚C.

## Flow cytometry

For activation assessment, the anti-CD69 FITC (Genetex GTX43516) was used. Cells were stained as previously described [15].

For redox measurements, cells were incubated for 15 minutes at 37˚C in staining buffer with either: 2.5 μM of MitoSOX Red Mitochondrial Superoxide Indicator (M36008 Thermo-Fisher Scientific); or 2 μM of Dihydroethidium (DHE) (D11347 ThermoFisher Scientific); or 5 μM of Carboxy-$H_2$DCFDA (C400 ThermoFisher Scientific). Cells were washed twice with staining buffer.

Flow cytometry was assessed on a FACScalibur cytometer using the CellQuest software. The software FlowJo (Tree Star, CA) was used for analysis.

## Statistical analysis

Three to nine biological replicates were used for all experiments presented. Data are presented as means and standard deviations. An unpaired Mann-Whitney test was used for comparison between samples, and the Wilcoxon test was used for comparison between paired samples. The p-value is presented for each comparison, statistically significant differences are indicated.

## Logical modeling

To model the T cell network, we first defined a *regulatory graph*, where each node represents a component of the signaling network. Most nodes are associated with Boolean variables, but some are allowed to take three levels of activity (0, 1 and 2) when biologically justified. Nodes are connected through arcs, which represent the regulatory influences between them, positive or negative. Next, we defined logical rules to determine the level of activity of each target node as a function of the levels of its regulators (S1 Table).

We used the GINsim software (v3.0.0b, [24] http://ginsim.org) to build the TCR-REDOX--Metabolism signaling model and to identify its stable states.

We used the MaBoSS software (v2.0,[26, 27]) to perform stochastic simulations of our logical model. These simulations use default parameters: 50'000 runs with identical up and down rates for all components.

All our model analyses are reproducible using an interactive Python notebook and virtual software environment, available together with the model file on GINsim repository (http://ginsim.org/node/229) [28].

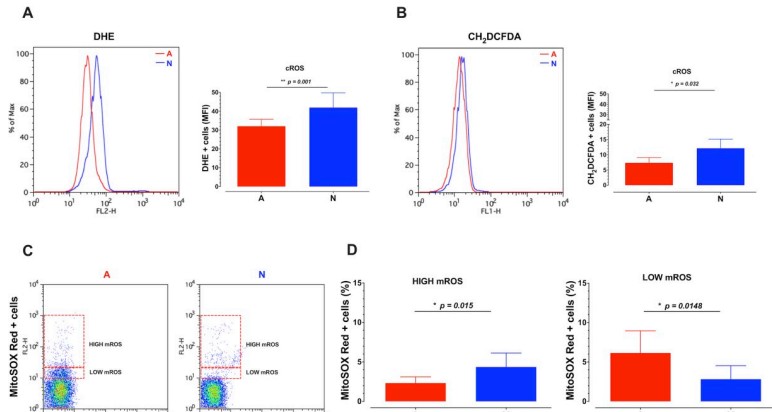

**Fig 1. Neonatal CD8⁺ T cells have higher basal cROS and mROS levels than their adult counterparts.** Purified human CD8⁺ T lymphocytes from neonate (N) or adult (A) donors were incubated with 2 μM of DHE (A), or 10 μM of Carboxy-$H_2$DCFDA (B) or 2.5 μM of the MitoSOX Red (C, and D) fluorescent dyes for 15 minutes. The fluorescence of the cells was assessed using a FACScalibur Flow Cytometer. The graphs show the fluorescence of the cells on the FL-2H channel (DHE, MitoSOX Red) or FL-1H channel ($CH_2$DCFDA). Five to nine samples per group are shown. An unpaired Mann-Whitney test was used for comparisons between samples from N and A. The p-value is presented for each comparison.

## Results

### Evaluation of ROS in neonatal and adult CD8⁺ T cells

Glycolytic enzymes are over-expressed in the neonatal CD8⁺ T cells, correlating with higher ROS production [18, 29, 30]. We first considered additional naïve CD8⁺ T cell samples, obtained as previously described [25] and a second ROS sensitive probe to corroborate the higher cROS levels in the neonatal cells (Fig 1A and 1B). We then measured mROS levels and identified two cell populations based on these (high vs low) (Fig 1C). We quantified the percentage of cells in the high or low mROS gates from neonatal or adult CD8⁺ T cells. We found that neonatal cells have a higher percentage of cells with high mROS, whereas adult cells are more frequent in the low mROS gate (Fig 1D).

Altogether these results demonstrate that a higher proportion of neonatal CD8⁺ T cells display high ROS levels, both in mitochondria and cytoplasm.

### T cell activation and mROS

T cell activation induces glycolysis, which could affect the production of ROS [31]. We thus evaluated the changes in ROS after TCR/CD28 cross-linking. Cell activation did not significantly change the levels of cROS (Fig 2A). In mitochondria, however, activation of adult cells led to a reduction in the proportion of low mROS cells, which could be due to the Warburg effect [32]. On the contrary, in the neonatal cells, stimulation increased the proportion of cells with high mROS (Fig 2A).

The proton channel Hv1 exports protons generated by NADPH-oxidase, which leads to the production of $H_2O_2$, contributing to a high level of cROS [33]. We measured the expression of Hv1 in neonatal and adult CD8⁺ T cells, before and after stimulation. In adult cells, Hv1 expression diminished after activation, while in those of neonates, activation induced Hv1 expression (Fig 2B).

To measure CD8⁺ T cell activation, we evaluated CD69 expression. Stimulation induced a higher expression of CD69⁺ cells in adult as compared to neonatal cells. Additionally, in the

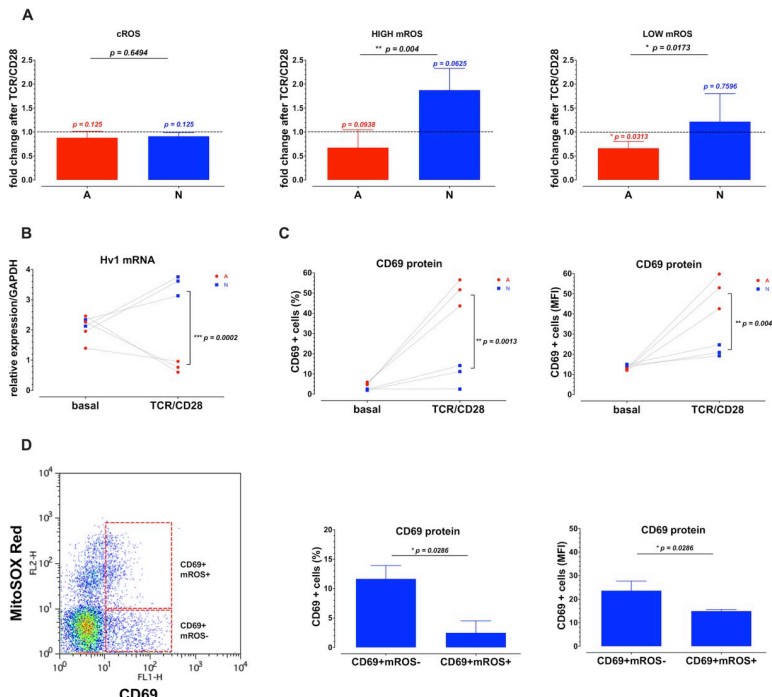

**Fig 2. The initial redox stress impairs the activation of neonatal human CD8$^+$ T cells upon TCR/CD28 stimulation.** Purified human CD8$^+$ T lymphocytes from neonate (N) or adult (A) donors were stimulated with anti-CD3/anti-CD28 antibodies. After 6 hours, the cells were collected for measurements. For the ROS evaluation, cells were incubated with 2 μM of DHE or 2.5 μM of the MitoSOX Red (A) fluorescent dyes for 15 minutes. The fluorescence of the cells was assessed using a FACScalibur Flow Cytometer. The graphs show the fluorescence or frequency of the cells after normalization to the basal levels of a total of 5 and 6 neonatal and adult cells' samples, respectively. (B) Total RNA was extracted from untreated or TCR/CD28 stimulated cells, using the TRIzol reagent. The Hv1 proton channel mRNA levels were evaluated using specific primers on a qPCR using the GAPDH mRNA levels as a reference gene. During our measurements, (C) The cells were washed and incubated with an anti-CD69 FITC fluorescent antibody for 30 mins the fluorescence or frequency of the cells was analyzed using a FACScalibur Flow Cytometer. (D) A MitoSOX Red and anti-CD69 FITC double staining was performed on CD8$^+$ T cells from neonates. The fluorescence or the frequencies of CD69$^+$mROS$^-$ and CD69$^+$mROS$^+$ populations is represented on the bar graphs. An unpaired Mann-Whitney test was used for comparisons between samples from neonatal (N) and adult (A) cells, considering four samples per group. The p-values between the bars denote the significance between adult and neonatal cells. On top of each bar, we show the significance between paired samples of stimulated N or A groups as compared with the non-stimulated cells. A Wilcoxon test was used for these comparisons.

high mROS gate of the neonatal cells, expression of CD69 was lower in comparison to the cells not producing mROS (Fig 2C and 2D).

## Modeling the interplay between T cell activation and mROS

To further understand the interplay between metabolism, ROS and T cell activation, we integrated our own data and literature information on interactions between metabolic pathways and cell signaling in our previous TCR logical model [34]. The resulting model shown in Fig 3 integrates 111 components and 244 interactions or arcs, of which 64 components and 168 interactions are new. This graph includes two inputs corresponding to TCR and CD28 signals. Second messengers, as well as mROS and cROS were also considered, together with key components of glycolysis, the citric acid cycle, the pathways of fatty acid synthesis and breakdown, of the oxidation of glutamine and pentoses, as well as the electron transport chain. Selected transcription factors link these pathways to output nodes representing cell responses, including Activation, Quiescence (dormant cells), Anergy (incomplete activation of transcription

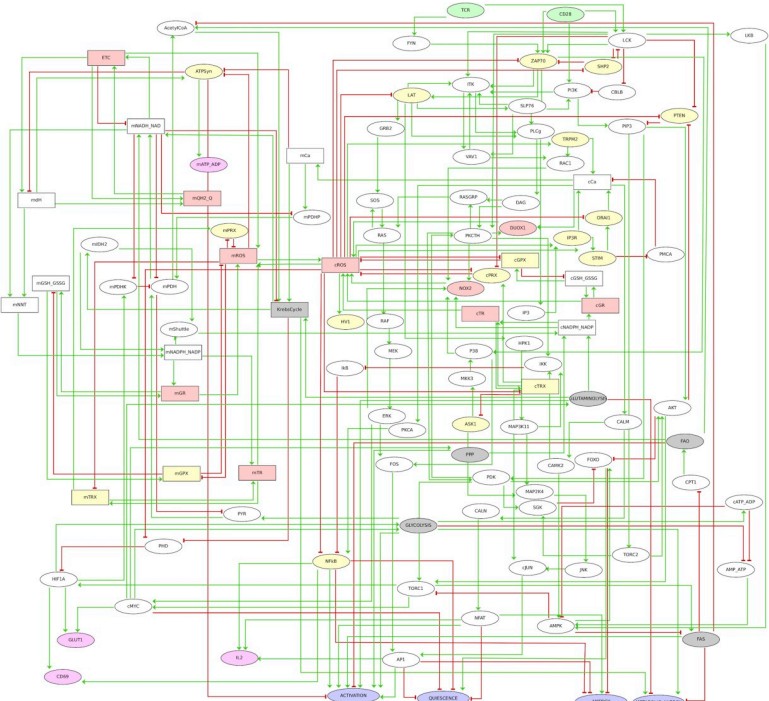

**Fig 3. Logical regulatory graph of the TCR-REDOX metabolism model.** The model was designed to study the interaction between the redox state and cellular metabolism and its influence on the activation of human CD8⁺ T lymphocytes. The nodes can be divided into three main categories: mitochondrial nodes (upper left), TCR signaling nodes (upper right), cytosol component nodes (center right and bottom right). A color code further denotes the type of node: green (input), gray (metabolic pathway), pink (ROS or ROS source), yellow (redox sensitive), magenta (output) and purple (phenotype). Elliptical shapes denote Boolean nodes (taking the values 0 or 1), while rectangular shapes denote multilevel (ternary) nodes (taking the values 0, 1 or 2). The arcs connecting the nodes represent positive (green) or negative (red) regulatory influences. In the case of multilevel regulatory nodes, the thresholds required for the activation of the different target nodes can be found in the model available online (http://ginsim.org/node/229).

factors) and Metabolic Anergy (aerobic glycolysis without oxidative mitochondrial metabolism). We also included IL-2 and CD69 as early activation markers. Logical rules specify how each component responds to incoming interactions. The model, including extensive annotations, is provided at the url: http://ginsim.org/node/229 (see also S1 Table).

To evaluate the predictive role of the model, we computed its stable states (Fig 4A, upper panel). In the absence of TCR or CD28 signal, cells remain quiescent or are kept in metabolic anergy. In the presence of both TCR and CD28 signals, cells could either turn anergic, with high ROS, or undergo activation.

Next, we enforced a fixed high ROS perturbation in order to compute the impact of a high ROS levels in neonatal CD8⁺ T cell activation. We introduced the perturbation either with mROS or cROS at the maximum functional levels and evaluated the outcome of T cell activation. The stable states under high ROS levels lead to two anergy states in which TCR or CD28 were engaged separately. When the full CD3/CD28 signals were engaged, the cells underwent an incomplete activation state, in which NFAT was activated but not NFκB, AP1 or IL-2 (Fig 4A). In this condition, anergy, metabolic anergy, or a combination of them are predicted (Fig 4A, middle panel).

High arginase levels have been reported in the serum of neonates, which could lead to lower glutamine levels. We thus computed the effect of a blockade of glutamine breakdown (Fig 4A, lower panel), which resulted in anergy, metabolic anergy and incomplete activation.

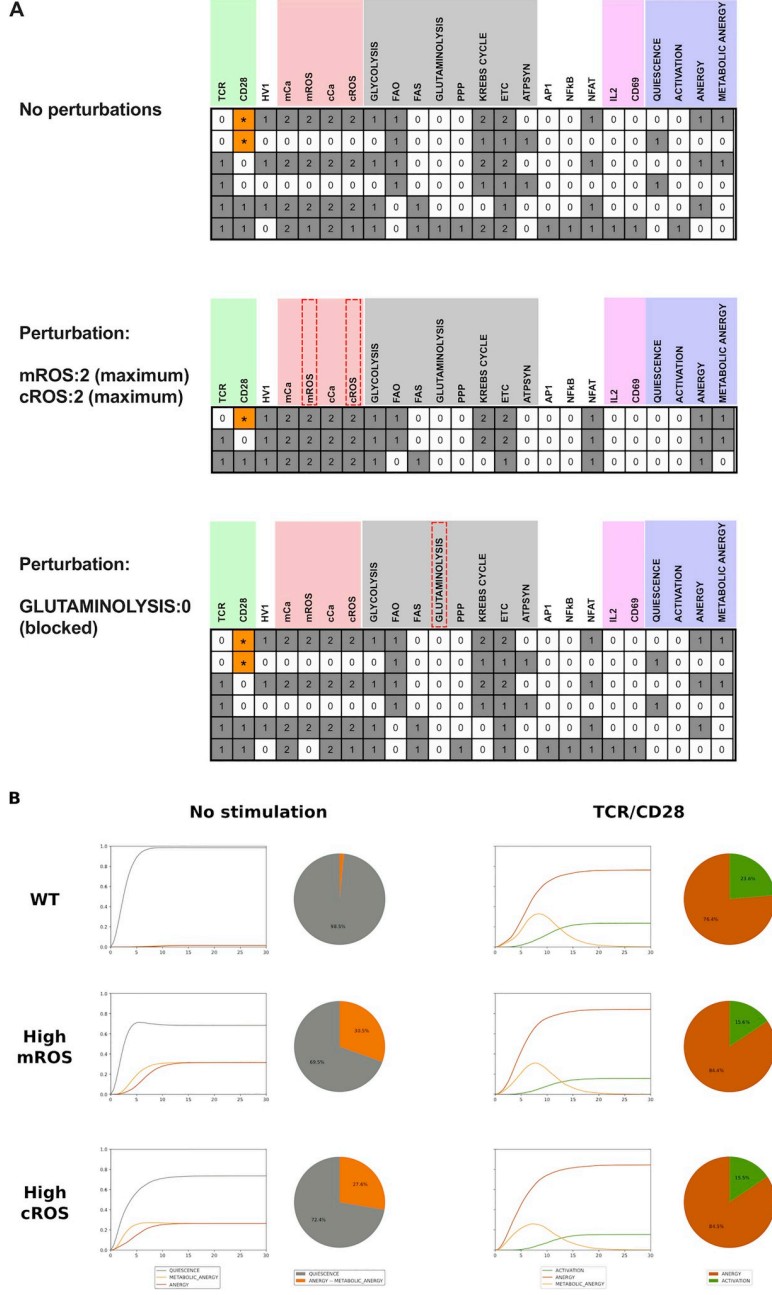

**Fig 4. The analysis of the logical model recapitulates the impact of metabolism on T cell activation.** (A)
Computation of the stable states on selected nodes of the model in three different scenarios (No perturbation; mROS: 2
or cROS: 2 and GLUTAMINOLYSIS: 0). Each row represents a single stable state. For each node, the white cells (0)
denote the lowest functional levels, while the gray cells denote intermediate (1) or maximum (2) levels of activity. We
further used the MaBoSS software to evaluate the reachability of the different phenotype nodes in the absence of
stimulation or upon TCR/CD28 stimulation (B). The pie charts represent the frequency of each phenotype under each
condition, while the time plots represent the evolution of each phenotype node over time.

Finally, using the MaBoSS tool, ([26, 27] https://maboss.curie.fr/), we performed stochastic
simulations to assess the reachability of these stable states in unstimulated (Fig 4B left) or
TCR/CD28 stimulated cells (Fig 4B right). In the non-stimulated (NS) scenario, the majority
of the cells (98%) remained quiescent, and only under 2% undergo metabolic anergy. After

activation, 23% of cells become active. With fixed activation of either mROS or cROS, 31% or 25% of cells undergo metabolic anergy, respectively. Under these conditions, only 15% of cells reach an active state.

In conclusion, our model thus predicts a dramatic effect of ROS status on T cell activation, reproducing our own data, as well as previous observations reviewed in [35].

## Discussion

Energy metabolism has a dramatic effect on immune cell homeostasis and response and is determinant for health and disease. Neonatal T cells have a unique metabolic and signaling profile, which results in a low response to stimulation.

Neonatal CD8⁺ T cells encompass a higher percentage of cells with a very high mROS levels, while low mROS cells are reduced in comparison with adults. Low levels of ROS promote T cell activation, particularly due to the inhibition of phosphatases and the potentiation of NFκB signaling. High levels of ROS are in contrast detrimental for cell activation [35].

One of the most important signaling hubs is intracellular $Ca^{2+}$. This second messenger is altered in neonatal lymphocytes. In mouse cells, particularly in CD4⁺ T cells, it was reported that calcium fluxes are increased in neonatal cells [20]. However, in human cells, it was reported that neonatal and adult CD8⁺ T cells have equivalent calcium fluxes [19]. Our model predicts that high ROS will increase cytosolic calcium, leading to the activation of NFAT and thus a preferential expression of IL-4 over that of pro-inflammatory cytokines [36]. This is in agreement with high IL-4 expression in neonatal cells and other conditions in which high ROS induces the expression of IL-4 [37, 38].

The high ROS levels of neonatal cells could be attributed to active glycolysis already in unstimulated cells [18] and/or to a low capacity of their mitochondria to control electrons along the Respiratory Chain, producing mROS [39]. Interestingly, cell activation induces Hv1 channel in neonatal but not adult cells, in which its expression is reduced. This channel expels the excess protons generated by the action of NADPH-oxidase. Recent evidence suggests that this channel is also located in the internal mitochondrial membrane, contributing to the production and modulation of mROS [40]. This suggest that neonatal cells cannot use the high levels of NADH produced by glycolysis. A low mitochondrial function of the neonatal cells has been reported [19]. If, as predicted by the model, high ROS induces a high level of calcium waves in neonatal cells, this might convert the mitochondria into calcium storage organelles, which could reduce their metabolic function [41].

Based on our model, we predict that high levels of ROS prevent the activation of the pentose phosphate pathway, fatty acid synthesis and glutamine breakdown, which are necessary for proper T cell activation. The microenvironment of the neonatal lymphocytes shows high levels of the enzyme arginase [42], which would limit Arginine, which in turn would affect the glutamine available for synthesis *de novo* of glutathione [31]. The correct balance of these metabolic activities is crucial for T cell function, as exemplified by infiltrating lymphocytes in a tumor microenvironment in which ROS, glutaminase and arginase contribute to lower the activation potential of immune cells [15–17, 22, 23].

The dynamic analysis of our model indicates that under high ROS levels, over 25% of T cells would be in metabolic anergy, thereby lowering their activation potential, which would tentatively protect newborns from excessive activation at birth, when confronted with many novel antigens.

Some limitations of this study need to be declared, however, in order to consider our findings under the proper light. First, the number of samples was limited although we obtained statistical significance for our results. Second, the T cell pool from cord blood samples has a

considerable amount of recent thymic emigrants, with reduced activation potential and tolerant features. Identifying these populations have been a challenge for CD8+ T cells, because of the lack of a bona-fide phenotypic marker. The reliable marker of recent thymus emigrants only identifies those of CD4+ Tcells [43]. Third, these experiments were performed *in vitro*, hence, the influence of other relevant cell types (e.g. dendritic cells and macrophages) in the surrounding microenvironment around CD8+ T cells in the redox signals could not be assessed.

In conclusion, the metabolic and redox profile of neonatal lymphocytes tentatively impairs their activation potential. This should be addressed in studies aiming at boosting neonatal immunity. In addition, our model could be useful in other situations, e.g. to identify the nodes that could be targeted in order boost T cell effector function in tumors.

## Supporting information

**S1 Table. Annotations for the TCR-REDOX-Metabolism model and specification of the logical rules.** This table has been generated using an export function of the software GINsim and lists the following information for each node of the model (first column):

- a series of database entry identifiers documenting the sources of information used to build the model (second column);

- the Boolean rules defined for each node; note that in the case of multilevel (ternary) nodes, two rules are specified, for values 2, and 1, respectively (third column, upper part of the cells); these rules combine literals (node names) with the standard Boolean operators NOT (denoted by the symbol !), AND (denoted by &), OR (denoted by |), and parentheses whenever required;

- textual annotations explicating the underlying modeling assumptions.
(DOC)

**S1 Fig.**
(TIFF)

## Acknowledgments

We thank Centro Estatal de la Transfusión Sanguínea (Morelos) and Hospital José G. Parres for access to the blood samples. We also thank the mothers of the babies participating on the study, together with all members of the Santana and Thieffry labs. We thank Prof. Chris. Pogson for the copyedit revision of the manuscript.

## Author Contributions

**Conceptualization:** Denis Thieffry, María Angélica Santana.

**Formal analysis:** José Antonio Sánchez-Villanueva, Otoniel Rodríguez-Jorge, Wassim Abou-Jaoudé, Denis Thieffry, María Angélica Santana.

**Investigation:** José Antonio Sánchez-Villanueva, Otoniel Rodríguez-Jorge, Oscar Ramírez-Pliego, Gabriela Rosas Salgado, Céline Hernandez, Aurélien Naldi.

**Methodology:** José Antonio Sánchez-Villanueva.

**Software:** Wassim Abou-Jaoudé, Céline Hernandez, Aurélien Naldi.

**Supervision:** Denis Thieffry, María Angélica Santana.

**Validation:** José Antonio Sánchez-Villanueva.

**Writing – original draft:** José Antonio Sánchez-Villanueva, Denis Thieffry, María Angélica Santana.

**Writing – review & editing:** Denis Thieffry, María Angélica Santana.

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
