## [Decision Letter · Decision Letter 0]

29 Oct 2019

PONE-D-19-24723

Contribution of ROS and metabolic status to  neonatal and adult CD8+T cell activation

PLOS ONE

Dear Dr. Santana,

Thank you for submitting your manuscript to PLOS ONE. After careful consideration, it was felt that it has merit but does not fully meet PLOS ONE’s publication criteria as it currently stands. Therefore, we invite you to submit an extensively revised version of the manuscript that addresses all the issues raised during the review process.

Please note that the revised version of the manuscript will be sent back to the original reviewers prior to final acceptance.

We would appreciate receiving your revised manuscript by Dec 13 2019 11:59PM. To enhance the reproducibility of your results, we recommend that if applicable you deposit your laboratory protocols in protocols.io, where a protocol can be assigned its own identifier (DOI) such that it can be cited independently in the future. For instructions see: http://journals.plos.org/plosone/s/submission-guidelines#loc-laboratory-protocols

We look forward to receiving your revised manuscript.

Yours Sincerely,

Lucienne Chatenoud

Academic Editor

PLOS ONE

Journal Requirements:

2. Thank you for your ethics statement : "Servicios de Salud Morelos, Comité de Etica en Investigación del Hospital General de Cuernavaca Dr. José G. Parres. CONBIOÉTICA-17-CEI-001-20160329

Secretaría de Salud. Dirección de Atención Médica Departamento Centro Estatal de la Transfusión Sanguínea Oficio No: SH/CETS/438/2016"

a) Please amend your current ethics statement to confirm that your named institutional review board or ethics committee specifically approved this study.

Reviewers' comments:

Reviewer's Responses to Questions

**Comments to the Author**

1. Is the manuscript technically sound, and do the data support the conclusions?

Reviewer #1: Partly

Reviewer #2: Yes

2. Has the statistical analysis been performed appropriately and rigorously? 

Reviewer #1: No

Reviewer #2: Yes

3. Have the authors made all data underlying the findings in their manuscript fully available?

Reviewer #1: Yes

Reviewer #2: Yes

4. Is the manuscript presented in an intelligible fashion and written in standard English?

Reviewer #1: Yes

Reviewer #2: No

5. Review Comments to the Author

Reviewer #1: While interesting major elements are missing from the paper and need to be added.

1. Important details are missing in the methods, particularly the number of cord blood and adult samples obtained. Were these from particular sexes? What was the time from the obtaining of the cord blood samples to their processing? Was this variable and did this affect the result. What were the age and sex of the adult donors? Was there a similar interval between taking of blood from adult donors and its processing? Was the cord blood from natural births or caesarians. Had the mothers received any drugs during labour that might have affected the T cell responses. Was any attempt made to obtain parallel blood from the mothers to compare to the cord blood as this would exclude any drug or hormonal confounders that might be affecting the cord blood T cells.

2. Fig 2 seems to suggest there were only 3 cord blood and 3 adult samples obtained? How do the authors know these responses are representative? As cord blood may not be representative of blood of young children due to prevailing maternal and birth stress factors, was any attempt made to obtain blood from young babies to show that these exhibited the same pattern as the cord blood samples?

3. In view of the above the authors need to add a paragraph at the end of the study describing shortcomings in the study design and cautions in interpretation of the data.

Reviewer #2: The manuscript by Sanchez-Villanueva et al sought to determine the correlation of reactive oxygen species (cytosol and mitochondria) against the known lower T cell activation profile of neonates. Specifically, they obtained cord blood or adult blood and stimulated with a polyclonal activator and then determined mROS and cROS in CD8 T cells. They found neonates consistently had high levels of mROS while adults had a mix of high and low mROS. They then built a logic model to depict the effects of activation on neonatal CD8 T cells.

The overall results of the study are quite interesting with respect to the mROS differences. There are a few issues that should be addressed to improve the manuscript:

1. A through reading of the manuscript to improve the wording would help the reader. There are few word choices and sentence mechanics all throughout the manuscript that make it a bit difficult to read.

2. In figure 2, the authors found a large difference in HV-1 expression between infants and adults. However, it appears that GADPH was used as the normalization control which is no longer appropriate. GADPH is so widely impacted by immunological processes (i.e. TNFa or IFNg mRNA transcripts bind to it) that it is not an appropriate housekeeping control. Their data may still be relevant, but they should ensure the rigor of their findings with another calibrator, especially when comparing neonates to adults where transcripts such as IFNg are widely different during activation

3. Comparisons of neonates to adults naïve T cells is not an apples to apples comparison. Neonates are loaded with recent thymic emigrants in blood that express lower T cell function that may enter the competitive pool for activation. Since the authors did not gate on these or remove them, they should at least bring up the topic of RTEs in the discussion.

4. The authors should at least discuss the high mROS they observed in the context of NFAT and IL-2. Both of these are lower in neonates, affect the activation profile even at 6 hours, and are inhibited by high mROS. Contrary, IL-4 which is supposedly higher in neonates is also inhibited by high mROS and thus this should all be in the discussion section.

5. Calcium influx increases mROS in T cells as determined in a number of prior studies. The authors cite reference 20 to indicate that influx may be high in infants. There are a number of studies that indicate that the influx may also be lower in neonates and that the increase is only in CD4 T cells (and non-Th1 cells at that). They authors should at least discuss this.

6. PLOS authors have the option to publish the peer review history of their article (what does this mean?). If published, this will include your full peer review and any attached files.

Reviewer #1: No

Reviewer #2: No

---

## [Author Response · Author response to Decision Letter 0]

8 Nov 2019

Responses to Editor and Reviewers:

We thank the editor and the two reviewers for their constructive criticism, which enabled us to greatly improve our manuscript. 

Please find enclosed below a point by point answer to all concerns:

Editor:

We have reviewed the manuscript, labeled the former and new modified file as asked by PLOS ONE

2. (a) Please amend your current ethics statement to confirm that your named institutional review board or ethics committee specifically approved this study.

We have changed the current ethics statement to: 

The collection of cord blood samples, with informed mothers’ consent was granted for this work by Comité de Etica en Investigación Hospital General de Cuernavaca Dr. José G. Parres. CONBIOÉTICA-17-CEI-001-20160329

The collection of adult blood was granted by the ethics committee of Secretaría de Salud. Dirección de Atención Médica Departamento Centro Estatal de laTransfusión Sanguínea Oficio No: SH/CETS/438/2016.

This is shown in lines 131-136

2. (b) Once you have amended this/these statement(s) in the Methods section of the manuscript, please add the same text to the “Ethics Statement” field of the submission form (via “Edit Submission”).

We have done that.

The revised manuscript has been reviewed by Prof. Christopher Ian Pogson, English born scientist, graduated in Cambridge, UK. Former head of the Biochemistry Department of Manchester University Scientist and Editor of The Biochemical Journal, now retired.

Reviewers Comments to the Author:

Reviewer # 1: While interesting major elements are missing from the paper and need to be added.

1. Important details are missing in the methods, particularly the number of cord blood and adult samples obtained. Were these from particular sexes? What was the time from the obtaining of the cord blood samples to their processing? Was this variable and did this affect the result. What were the age and sex of the adult donors? Was there a similar interval between taking of blood from adult donors and its processing? Was the cord blood from natural births or caesarians. Had the mothers received any drugs during labour that might have affected the T cell responses. Was any attempt made to obtain parallel blood from the mothers to compare to the cord blood as this would exclude any drug or hormonal confounders that might be affecting the cord blood T cells.

For each comparison the minimum number of samples included in the neonate vs adult groups was three. For most cases the number of samples was five. The total number of samples used for this study was 30 samples for neonatal CD8+T cells and 30 samples of adult cells. Because we were using purified naïve cells, only 3 to 6 million CD8+ T cells were obtained from cord blood, and 5 to 10 million from adult samples. No eligibility criteria were defined regarding the donors’ gender, this parameter was distributed randomly across the samples, and an equivalent number of male and female donors was used for neonatal and adult cells. No differences related to gender were detected. For adult samples, the ages of the donors were in the range between 26 and 40 years. All cord blood samples were obtained from vaginal deliveries, no drugs known to affect T cells responses were administered to the mothers during labor, mostly because of budget limitations in the hospital. All the samples were immediately processed after collection. This information has been included in the Materials and Methods section of the revised manuscript, lines: 130-150. We did not collect blood from mothers because we only obtained permission for noninvasive procedures.

2. Fig 2 seems to suggest there were only 3 cord blood and 3 adult samples obtained? How do the authors know these responses are representative? As cord blood may not be representative of blood of young children due to prevailing maternal and birth stress factors, was any attempt made to obtain blood from young babies to show that these exhibited the same pattern as the cord blood samples?

In Fig 2A, we obtained 5 adult cells samples and 5 neonatal cell samples, in Figs 2B and 2C we obtained only 3 adult samples and 3 neonatal samples. For Fig 2D, we captured 4 neonatal blood samples. We made more evaluations of CD69 protein with the same results. Other members in the lab have also measured this protein with similar results, but together with ROS measurements, we made 3 evaluations with statistically significant results. The data shown in Fig 2 are consistent with a high ROS levels in the neonatal cells and come from independent biological samples (because of the limitations on the number of cells), thus strengthening our conclusions. Mothers in Hospital Parres do not receive anesthetics, hormones or drugs during delivery, unless they go into caesarian-section. All samples were captured from vaginal deliveries, with no treatment whatsoever. The number of CD8+ T cells for about 40 ml of cord blood is about 3 and 5 million cells. It is almost impossible to obtain CD8+T cells from infants later in life. In addition, the amount of blood that could be authorized for young infants’ samples is 1 mL. Care was taken to obtain the blood immediately after delivery and before placenta expulsion, to have a sample more representative of the infant blood. Cells were put in culture medium during the purification period of two days, which arrests the cells in resting conditions. We have compared cord blood from vaginal deliveries versus programmed caesarian sections and these proved to be largely similar. Although published reports claim that cord blood is not representative of the infant blood, these do not state how they obtained the samples (Olin et al. (2018). Cell 174 (5): 1277-92). Note that most labs are only allowed to get cord blood samples from arterial blood of expelled placentas, in which the blood started the coagulation process and both serum and cell types are altered because they get trapped in the clot. In contrast, in our study, cord blood was sampled immediately after delivery and before placenta expulsion.

3. In view of the above the authors need to add a paragraph at the end of the study describing shortcomings in the study design and cautions in interpretation of the data.

This has been included in the revised manuscript, lines 510-561.

Reviewer # 2: The overall results of the study are quite interesting with respect to the mROS differences. There are a few issues that should be addressed to improve the manuscript:

1. A through reading of the manuscript to improve the wording would help the reader. There are few word choices and sentence mechanics all throughout the manuscript that make it a bit difficult to read.

The revised manuscript has been reviewed by Prof. Christopher Ian Pogson, English born scientist, graduated in Cambridge, UK. Former head of the Biochemistry Department of Manchester University and Editor of The Biochemical Journal now retired. 

2. In figure 2, the authors found a large difference in HV-1 expression between infants and adults. However, it appears that GADPH was used as the normalization control which is no longer appropriate. GADPH is so widely impacted by immunological processes (i.e. TNFa or IFNg mRNA transcripts bind to it) that it is not an appropriate housekeeping control. Their data may still be relevant, but they should ensure the rigor of their findings with another calibrator, especially when comparing neonates to adults where transcripts such as IFNg are widely different during activation.

A set of experiments were performed with the aim of selecting the proper reference gene for qPCR comparisons. We are enclosing below a table with information regarding the quality of the primers.

The relative expression of these genes was evaluated in Peripheral Blood Mononuclear Cells (PBMC) and Cord Blood Mononuclear Cells (CBMC) samples at 0, 3 and 6 hours after anti-CD3/anti-CD28 in vitro stimulation. The gene GAPDH was the one selected due to the stability of its expression levels across the stimulation period (Fig 1A).

In Fig 1B, we present the relative expression of GAPDH mRNA upon anti-CD3/anti-CD28 in vitro stimulation on purified CD8+T cells from neonates or adults. No differences were detected due to the effect of the TCR stimulus.

Fig 1. Evaluation and selection of the internal reference gene for qPCR comparisons. Total RNA was extracted from untreated or TCR/CD28 stimulated cells, using the TRIzol reagent. The GAPDH, B2M and ACTB mRNA levels were evaluated using specific primers on a qPCR. (A) PBMC or CBMC were evaluated. (B) Purified CD8+ T cells from neonates (N) or adults (A) were evaluated. A paired two-tailed t test was used for comparison between samples.

3. Comparisons of neonates to adults naïve T cells is not an apples to apples comparison. Neonates are loaded with recent thymic emigrants in blood that express lower T cell function that may enter the competitive pool for activation. Since the authors did not gate on these or remove them, they should at least bring up the topic of RTEs in the discussion.

The T cell pool from cord blood samples has a considerable amount of recent thymic emigrants, with reduced activation potential and tolerance features. Identifying these populations have been a challenge for CD8+T cells, because of the lack of a bona-fide phenotypic marker. The reliable marker of recent thymus emigrants only identifies CD4+ T cells. We discuss this point in the revised manuscript, lines 512-516.

4. The authors should at least discuss the high mROS they observed in the context of NFAT and IL-2. Both of these are lower in neonates, affect the activation profile even at 6 hours, and are inhibited by high mROS. Contrary, IL-4 which is supposedly higher in neonates is also inhibited by high mROS and thus this should all be in the discussion section.

The cases of NFAT and IL2 are now further commented in the revised manuscript, lines: 410-417, 467-470. For IL-4, as suggested by the reviewer, we included the impact of mROS on IL-4 expression in the discussion 567-470.

5. Calcium influx increases mROS in T cells as determined in a number of prior studies. The authors cite reference 20 to indicate that influx may be high in infants. There are a number of studies that indicate that the influx may also be lower in neonates and that the increase is only in CD4 T cells (and non-Th1 cells at that). They authors should at least discuss this.

We added a paragraph addressing this point in the revised manuscript, lines: 463-470.

---

## [Decision Letter · Decision Letter 1]

26 Nov 2019

Contribution of ROS and metabolic status to neonatal and adult CD8+ T cell activation

PONE-D-19-24723R1

Dear Dr. Santana,

We are pleased to inform you that the revised version of your manuscript has been judged scientifically suitable for publication and will be formally accepted for publication once it complies with all outstanding technical requirements.

Yours Sincerely,

Lucienne Chatenoud

Academic Editor

PLOS ONE

Additional Editor Comments (optional):

Reviewers' comments:

Reviewer's Responses to Questions

**Comments to the Author**

1. If the authors have adequately addressed your comments raised in a previous round of review and you feel that this manuscript is now acceptable for publication, you may indicate that here to bypass the “Comments to the Author” section, enter your conflict of interest statement in the “Confidential to Editor” section, and submit your "Accept" recommendation.

Reviewer #2: All comments have been addressed

2. Is the manuscript technically sound, and do the data support the conclusions?

Reviewer #2: (No Response)

3. Has the statistical analysis been performed appropriately and rigorously? 

Reviewer #2: (No Response)

4. Have the authors made all data underlying the findings in their manuscript fully available?

Reviewer #2: (No Response)

5. Is the manuscript presented in an intelligible fashion and written in standard English?

Reviewer #2: (No Response)

6. Review Comments to the Author

Reviewer #2: (No Response)

7. PLOS authors have the option to publish the peer review history of their article (what does this mean?). If published, this will include your full peer review and any attached files.

Reviewer #2: No

---

## [Editor Report · Acceptance letter]

4 Dec 2019

PONE-D-19-24723R1 

Contribution of ROS and metabolic status to neonatal and adult CD8+ T cell activation 

Dear Dr. Santana:

I am pleased to inform you that your manuscript has been deemed suitable for publication in PLOS ONE. Congratulations! Your manuscript is now with our production department. 

With kind regards,

on behalf of

Prof. Lucienne Chatenoud 

Academic Editor

PLOS ONE